# Identification and characterization of specific motifs in effector proteins of plant parasites using MOnSTER

Giulia Calia[1,2,3,8], Paola Porracciolo[3,4,8], Yongpan Chen[3,7], Djampa Kozlowski[3,4], Hannes Schuler[1,5], Alessandro Cestaro[2,6], Michaël Quentin[3], Bruno Favery[3], Etienne G. J. Danchin [3,9] & Silvia Bottini [3,4,9] ✉

Plant pathogens cause billions of dollars of crop loss every year and are a major threat to global food security. Identifying and characterizing pathogens effectors is crucial towards their improved control. Because of their poor sequence conservation, effector identification is challenging, and current methods generate too many candidates without indication for prioritizing experimental studies. In most phyla, effectors contain specific sequence motifs which influence their localization and targets in the plant. Therefore, there is an urgent need to develop bioinformatics tools tailored for pathogen effectors. To circumvent these limitations, we have developed MOnSTER a specific tool that identifies clusters of motifs of protein sequences (CLUMPs). MOnSTER can be fed with motifs identified by de novo tools or from databases such as Pfam and InterProScan. The advantage of MOnSTER is the reduction of motif redundancy by clustering them and associating a score. This score encompasses the physicochemical properties of AAs and the motif occurrences. We built up our method to identify discriminant CLUMPs in oomycetes effectors. Consequently, we applied MOnSTER on plant parasitic nematodes and identified six CLUMPs in about 60% of the known nematode candidate parasitism proteins. Furthermore, we found co-occurrences of CLUMPs with protein domains important for invasion and pathogenicity. The potentiality of this tool goes beyond the effector characterization and can be used to easily cluster motifs and calculate the CLUMP-score on any set of protein sequences.

Plant pathogens are a major threat to global food security. To cause the infection, pathogenic organisms secrete effector proteins that promote colonization of the host plant by overcoming the physical barriers of plant cell walls, suppressing or evading immune perception, and deriving nutrients from host tissues[1]. Therefore, identifying and characterizing pathogens effectors is crucial towards understanding how they manipulate the plant and better combat them. Effector proteins are often specific to pathogens and essential for causing plant pathology, constituting targets of choice for the development of cleaner and more specific control methods[2–4]. Because of their poor sequence conservation, effector identification among the set of predicted proteins from the genome (proteome) is challenging and current methods generate too many candidates without further indication for prioritizing experimental studies. Classically, effector proteins are indirectly identified among the predicted secretome based on the presence of a signal peptide for secretion and a lack of transmembrane region[5,6]. However, these criteria alone suffer from two main limitations. On one side, the secretome comprises many proteins that are not effectors, on the other side some known effectors do not possess signal peptides for secretion. In most phyla, effectors contain specific sequence motifs which target host proteins with distinct roles in the infection process and control virulence[7]. The best-studied example is effectors secreted via the type III secretion system (T3SS) class of Gram-negative bacterial pathogens which are characterized by a specific motif/domain conferring a repertoire of molecular determinants with important roles during infection[8,9]. However, these features are not conserved in other bacteria. Indeed, gram-positive pathogens and certain phloem- and xylem-colonizers, such as *Candidatus liberibacter* and *Xylella*

[1]Free University of Bolzano, Faculty of Agricultural Environmental and Food Science, Bolzano, Italy. [2]Fondazione Edmund Mach, Research and Innovation Centre, San Michele all'Adige, Italy. [3]INRAE, Université Côte d'Azur, CNRS, Institut Sophia Agrobiotech, Sophia-Antipolis, France. [4]Université Côte d'Azur, Center of Modeling, Simulation and Interactions, Nice, France. [5]Free University of Bolzano, Competence Centre for Plant Health, Bolzano, Italy. [6]Institute of Biomembranes, Bioenergetics and Molecular Biotechnologies (IBIOM), National Research Council (CNR), Bari, Italy. [7]Present address: Department of Plant Pathology, China Agricultural University, Beijing, China. [8]These authors contributed equally: Giulia Calia, Paola Porracciolo. [9]These authors jointly supervised this work: Etienne G.J. Danchin, Silvia Bottini. ✉e-mail: silvia.bottini@inrae.fr

spp., do not encode the T3SS. In these bacteria, effector delivery is dependent on the presence of the N-terminal signal peptide, which is required for protein secretion[10]. In fungi, often effectors are small in size and present cysteine-rich sequences[11]. Another well-characterized example is the effectors of the oomycetes pathogens. Oomycetes are eukaryotic filamentous and heterotrophic microorganisms among which, more than 60% of them parasitize plants[12]. Well-known plant pathogens in oomycetes include late blight of potato, sudden oak death, root rot agents (*Phytophthora* species), and downy mildew *Peronospora* and *Bremia* species[13,14]. These pathogens code for two notable classes of effector proteins RxLR and Crinkler (CRN), that can be predicted by the occurrence of the related motifs, RxLR, -dEER and LxLFLAK-HVLVxxP in the N-terminal region downstream the signal peptide[15–17].

Although for some plant pathogens such as oomycetes, effectors have been studied extensively and characteristics motifs have been identified[18,19], research on plant-parasitic nematode (PPN) effectors did not identify any consensus motif, conserved across multiple species. The most economically important PPNs are the sedentary Root-Knot Nematodes (RKNs) and cyst nematodes[20]. These sedentary parasites induce the formation of a feeding structure that serves as a constant food source for the nematode. Other PPNs are migratory and a whole spectrum of variations exists between endo and ecto parasites, with semi-endoparasites an intermediate between the two extremes[21]. The different lifestyles of PPNs are expected to be reflected in their secretions, which presumably contain effectors with different functions according to the nematode's specific needs, thus presenting a high variety of characteristic motifs complicating their identification.

A first step toward the identification of motif characteristics of RKN effectors was performed by Vens et al.[22]. The authors developed a bioinformatic tool called MERCI to identify motifs with high occurrences in a positive dataset (known effector sequences) and absent in the negative one (non-effector sequences). MERCI uses a graph-based approach incorporating physicochemical features of the amino acids composing protein sequences. By analysing the known effector sequences of the RKN species *Meloidogyne incognita*, one of the most important known crop pathogens among all[23], they identified 4 motifs. However, at the time of their publication, very few genomes for RKN species were available, and the study was therefore conducted on one single RKN species. Furthermore, the genome used at that time was later shown to be partially incomplete[24]. These limitations prevent the generalization of the previous findings. Da Rocha et al. identified a *cis*-regulatory promoter motif (Mel-DOG box) characteristic of dorsal gland effectors[25]. Recently, Rocha et al. used this motif combined with other criteria to select new putative effectors and validated 14 new dorsal gland-specific candidate effectors expressed in adult females[26]. Although all these studies have contributed to enlarging the list of known effectors, a global characterization of their properties is still missing. Therefore, there is an urgent need for improvements in the study of PPN effector sequences' properties and motif research.

By taking advantage of the multitude of proteomes available nowadays for several PPN, we developed a comprehensive motif mining analysis to identify characteristic motifs of candidate parasitism protein sequences of these species. Sequence motifs are usually of constant short size and are often repeated and conserved. Typically, motifs conform to a particular sequence pattern, where certain positions can be constrained to a specific amino acid, whereas others are not[27]. This confers a high degeneration of the motifs yielding a huge list of non-redundant motif sequences and consequently, some motifs that are not characteristics of effector sequences only[28]. Furthermore, different amino acids (AAs) can have similar physicochemical properties, thus different motif sequences can share similar properties. However, most available motif discovery tools do not consider these properties. To circumvent these limitations, we have developed MOnSTER a tool that identifies *clu*sters of *m*otifs of *p*rotein *s*equences (CLUMP) and associates a score to each CLUMP. This score encompasses the physicochemical properties of AAs and the motif occurrences. Overall, one of the key advantages of MOnSTER is that it reduces the redundancy of motifs found by de novo tools. Furthermore, already known motifs available in publicly available databases such as Pfam[29] and/or InterProScan[30] can also be used as input of MOnSTER to identify discriminant CLUMPs.

We built up our method to identify discriminant CLUMPs in 1743 candidate parasitism proteins of plant-pathogenic oomycetes. We showed the reliability of MOnSTER by identifying 5 CLUMPs that correspond to the known motifs: RxLR, -dEER and LxLFLAK-HVLVxxP. After this proof of concept, we applied MOnSTER on PPN effector proteins and identified peculiar motifs in their sequences at an unprecedented level. We selected a set of 4395 protein sequences from 13 PPN species belonging to the genera *Meloidogyne, Globodera, Heterodera, Radopholus* and *Bursaphelenchus*. We identified six CLUMPs present in 60% of the known effectors (positive dataset). Of note these CLUMPs were found in only 5% of the sequences of the negative datasets, thus highlighting the enrichment of the identified motifs in effector sequences. Furthermore, we found a specific co-occurrence of at least two CLUMPs in PPN candidate parasitism protein sequences bearing protein domains important for invasion and pathogenicity.

The potentiality of this tool goes behind the candidate parasitism proteins and can be used to easily cluster motifs and calculate the CLUMPs score on any set of protein sequences. Furthermore, we also provide a unique scoring system capable of measuring the physicochemical properties of motifs grouped in CLUMPs and a motif alignment algorithm to better explore chemical-physical properties within the CLUMPs. MOnSTER is freely available at https://github.com/Plant-Net/MOnSTER_PROMOCA.git[31].

## Results and discussion

### MOnSTER identified five CLUMPs containing known motifs characteristics of oomycetes effector protein sequences

Characteristic motifs of oomycetes effector proteins are well-known in the literature, such as RxLR, -dEER and LxLFLAK-HVLVxxP[15]. Thus, we reasoned to apply our tool, MOnSTER, on oomycetes effectors to test its ability to recover well-characterized motifs. We compiled a set of 4752 oomycetes proteins, comprising 1743 effectors and 3009 non-effectors, from five oomycetes species. We performed motif discovery on this set of proteins using MERCI and STREME and we identified 265 significantly enriched motifs (see the "Methods" section for further details). Then we fed MOnSTER with these motifs and we obtained 11 CLUMPs (Supplementary Table 1), employing the Davis–Bouldin score as a criterion to cut the tree. By selecting CLUMPs having a MOnSTER score greater than the median of the overall scores we identified six CLUMPs (CLUMP7, 4, 10, 6, 2 and 9), the first five best-scoring CLUMPs, according to the MOnSTER score, correspond to the known motifs (Fig. 1). In Fig. 2 we can also observe that the motifs are respectively grouped in two clades, the two characteristics motifs of CRN-effectors (LxLFLAK and HVLVxxP), form a separate subclade on the right, while the RxLR and -dEER motifs fall into the left clade, resembling the family distinction of effectors to which they belong. More precisely, RxLR motifs are divided into two different CLUMPs; CLUMP6 containing only RYLR and RFLR motifs, and CLUMP10, containing other RxLR motifs and included in the same sub-clade of the dEER motif (CLUMP2). The last best-scoring CLUMP contains no known motifs, perhaps suggesting a putative motif for oomycetes effectors to investigate. Since oomycetes effectors characterization is not in the scope of this article, we did not consider this last CLUMP for further analysis. In support of that, CLUMPs 7, 4, 10, 6 and 2 are present in 1205/1743 effectors (~70% of the sequences in the positive dataset), while in combination with the last best scoring CLUMP (CLUMP9) only two more sequences can be detected.

Thus, we investigated the occurrences and co-occurrences of the five selected CLUMPs in oomycetes effectors and non-effectors (Supplementary Fig. 1). For the effectors we deeply analysed the two distinct families; in total we found that 68% of the RxLR-effectors in the positive dataset contain the motifs in CLUMPs associated with the RxLR motif (CLUMP10, 6 and 2). In particular, CLUMP10 and 6 are present alone in 41% of the RxLR-effectors (1238/1743 RxLR-effectors), while 19% of the RxLR-effectors contained the co-occurrence of these CLUMPs with the CLUMPs representing the dEER

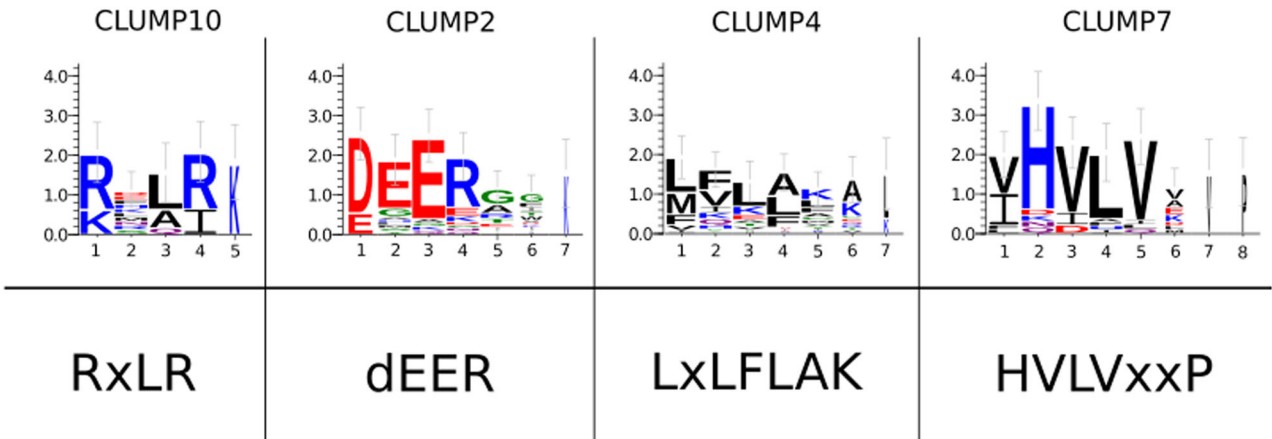

**Fig. 1 | Motif logos of CLUMPs compared to the target motifs.** Upper-panel: alignments of motifs in the respective CLUMP are produced by PROMOCA, and then the aligned motif sequences are used to produce the logos with WebLogo3. The x-axis represents the AA position in the motif, while the y-axis represents log-transformed frequencies translated into bits of information. Lower-panel: characteristic motifs of oomycetes effectors families from literature.

motif (CLUMP2). This reflects the importance of the RxLR motifs in the effector sequences and the role of the attached dEER[32]. On the other hand, the co-occurrence of CLUMPs specific for LxLFLAK and HVLVxxP (CLUMP7 and 4), in CRN-effector sequences accounts for 67% of the relative sequences in the positive dataset (377/1743). The high co-occurrences rate of CLUMP7 and 4 is strongly in agreement with the presence of LxLFLAK and HVLVxxP motif marking the beginning and the end of the DWL-domain in the Crinkler-effector family[33]. For the negative dataset, instead, only 15% of the sequences show the presence of CLUMP-motifs with a huge decrease in CLUMPs co-occurrences. Overall co-occurrences, indeed, are present in around 30% of positive sequences and in 1% of negative ones.

Previous research showed that the motifs characteristics of oomycetes effectors have strong sequence position preferences[34–36]. Thus, we plotted the CLUMPs occurrences in the positive versus negative dataset (Supplementary Fig. 2). Indeed, we can observe that the CLUMPs are concentrated at the beginning of the sequence in positive sequences and conversely spread around the sequence of negative dataset proteins. More precisely the five most interesting CLUMPs are condensed in the first 40% of the sequence with a higher preference at the very beginning and around 30% of the sequence, probably corresponding to the N-terminal of the protein in which the target motifs lie.

Altogether these results highlight the ability of MOnSTER to identify CLUMPs containing biologically relevant motifs.

## MOnSTER allowed us to identify six CLUMPs characteristics of nematode candidate parasitism proteins

The application of MOnSTER of the oomycetes effectors served as a proof of concept of our methodology. Thus, we moved to the characterization of nematode candidate parasitism sequences for which no characteristic motifs have been identified yet. We collected a set of 4395 proteins, including 546 well-known candidate parasitism proteins and 3849 proteins in the negative dataset, coming from 13 nematode species. By running motif discovery analysis as for the previous dataset, we found 269 motifs enriched in the candidate parasitism protein sequences. By applying MOnSTER with the previous configuration, the 269 input motifs were grouped into 11 CLUMPs. Six best-scoring CLUMPs were selected using the median as the threshold (Supplementary Table 2). Similar to the oomycetes results, we observe two main clades (Fig. 3): the second and the third best scoring ones (CLUMP2 and 5, respectively) form a single clade while the other selected CLUMPs (CLUMP1, 3, 7 and 10) are distributed in the bigger clade with the non-selected ones. Overall, we found at least one occurrence of one of the six CLUMPs in almost 60% of sequences from the positive dataset compared to 5% of sequences from the negative.

Then we investigated the presence of the six CLUMPs in each of the 13 PPN species present in the dataset. Figure 4 shows the abundance of the six best-scoring CLUMPs in the species according to their phylogeny tree. The first three species are the most represented in the positive dataset. Interestingly, very distant species show similar CLUMPs frequencies thus suggesting that they might share common characteristics at the sequence level for accomplishing similar functions. Furthermore, we could identify characteristic CLUMPs also for species represented in the dataset with very few sequences reinforcing the previous observation. Overall, this analysis suggests that CLUMPs might be associated with the functional properties of PPN nematodes.

Finally, we focused on the positional sequence preferences of CLUMPs in candidate parasitism protein sequences (Supplementary Fig. 3). In general, we observe a difference in the position preferences of the best-scoring CLUMPs between positive and negative dataset sequences. The six CLUMPs tend to occur more frequently in the middle of the sequences in candidate parasitism proteins (positive dataset), with more abundance in central (around 50% of the sequence) and terminal (around 70%), positions. The same CLUMPs are rare in the central position of the negative dataset protein sequences (negative dataset). Contrary to the properties of oomycetes effectors, whose characteristics CLUMPs occur mainly at the beginning of the sequence, PPN candidate parasitism proteins showed a different pattern of occurrences, privileging a central—C terminal occurrence.

## Co-occurrences of different CLUMPs are associated with functional protein domains

We investigated the co-occurrence patterns of CLUMPs in the PPNs candidate parasitism protein sequences (all possible combinations of co-occurrences are reported in Supplementary Fig. 4). Overall, we notice that CLUMPs tend to co-occur more frequently in the sequences of the positive dataset than in the negative ones, despite the positive set being smaller than the negative one. 30% of candidate parasitism protein sequences show co-occurrences of the six selected CLUMPs, while in the sequences from the negative dataset, co-occurrences, are present in less than 1% of the sequences. As observed for oomycetes, some CLUMPs tend to be present alone, while others tend to co-occur with specific CLUMPs. This suggests that different classes of nematode candidate parasitism proteins might exist, similar to the oomycetes effectors. Interestingly, among the 311 candidate parasitism proteins bearing at least one occurrence of one of the six selected CLUMPs, 72 do not have a predicted signal peptide, consisting of 55% of the proteins in the positive dataset not having the signal peptide. Of note, this is similar to the percentage of proteins bearing both the CLUMPs and the signal peptide, suggesting that CLUMPs characterize sequence properties beyond the type of secretion. Furthermore, similar patterns of

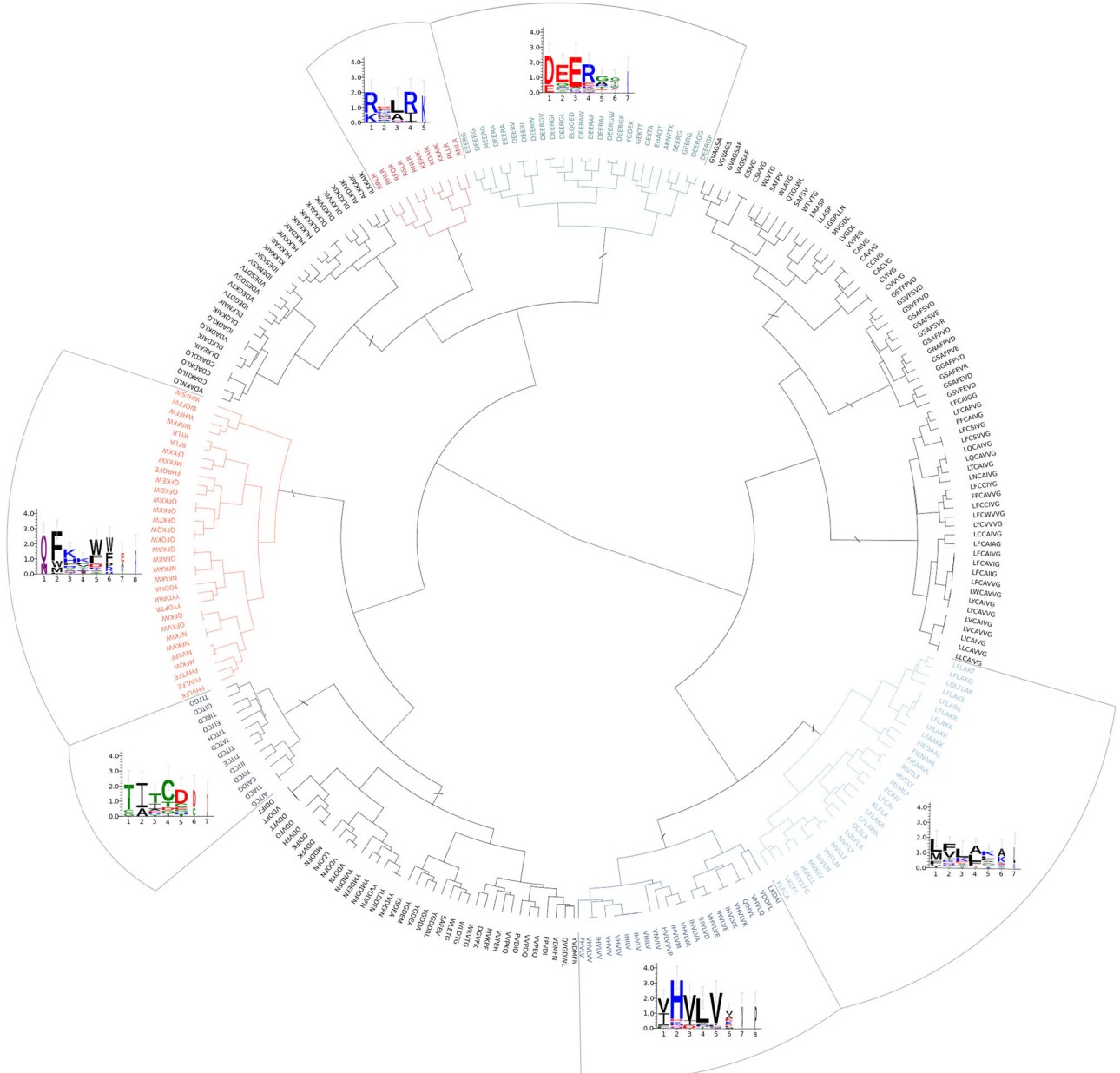

**Fig. 2 | Dendrogram of CLUMPs in Oomycetes.** 11 CLUMPs produced by MOnSTER (indicated with "/" sign). The coloured ones are those selected as best-scoring CLUMPs after MOnSTER-score calculation. Each best-scoring CLUMP is associated the motif logo; the alignment of the motif in each CLUMP is produced by PROMOCA and then WebLogo 3 is used to produce the image (the *x*-axis shows the AA position of the motif and the *y*-axis represents the log-transformed frequency of each AA in terms of bits of information).

co-occurrences of CLUMPs in candidate parasitism proteins bearing or not the signal peptide are observed with slightly higher co-occurrences presence in the sequences not having the signal peptide (Supplementary Fig. 5). Importantly, there is no relationship between the sequence length and the number of co-occurrences possibly suggesting a functional role for CLUMPs co-occurrences (Supplementary Fig. 6).

To inspect further a putative functional role of CLUMPs in candidate parasitism protein sequences, we queried the sequences having at least one CLUMP or co-occurrence of multiple CLUMPs against several protein domain databases (see the sections Methods and Results in Fig. 5 and Supplementary Data 1, sheet 1.5). Among the 311 candidate parasitism protein sequences bearing at least one occurrence of at least one of the six CLUMPs, 84 also have at least an occurrence of a known protein domain. The most recurrent hits are the coil domain, intrinsically disordered domain and the presence of the signal peptide (SP) followed by the pectate lyase

domain, glycosyl hydrolase family 5, Stichodactyla toxin (ShK) domain, 14-3-3 family and cysteine-rich domain. Importantly, none of these domains was also found in the sequences from the negative dataset bearing at least one occurrence of at least one of the six CLUMPs. Interestingly, we observe the almost exclusive association between CLUMPs and functional domains, mainly when multiple CLUMPs co-occur in candidate parasitism protein sequences.

The strongest association that we observe is between the co-occurrences of CLUMPs 7 and 10 and the glycosyl hydrolase family 5 domain on one hand and the co-occurrences of CLUMPs 3, 7, 10 and the cysteine-rich domain, on the other hand. Specifically, all 23 candidate parasitism protein sequences containing the co-occurrences of CLUMP 7 and 10 bear also the glycosyl hydrolase family 5 domain. By inspecting the position of CLUMPs occurrences within the sequences, we observed that the two CLUMPs are flanking the domain: CLUMP7 is consistently present at

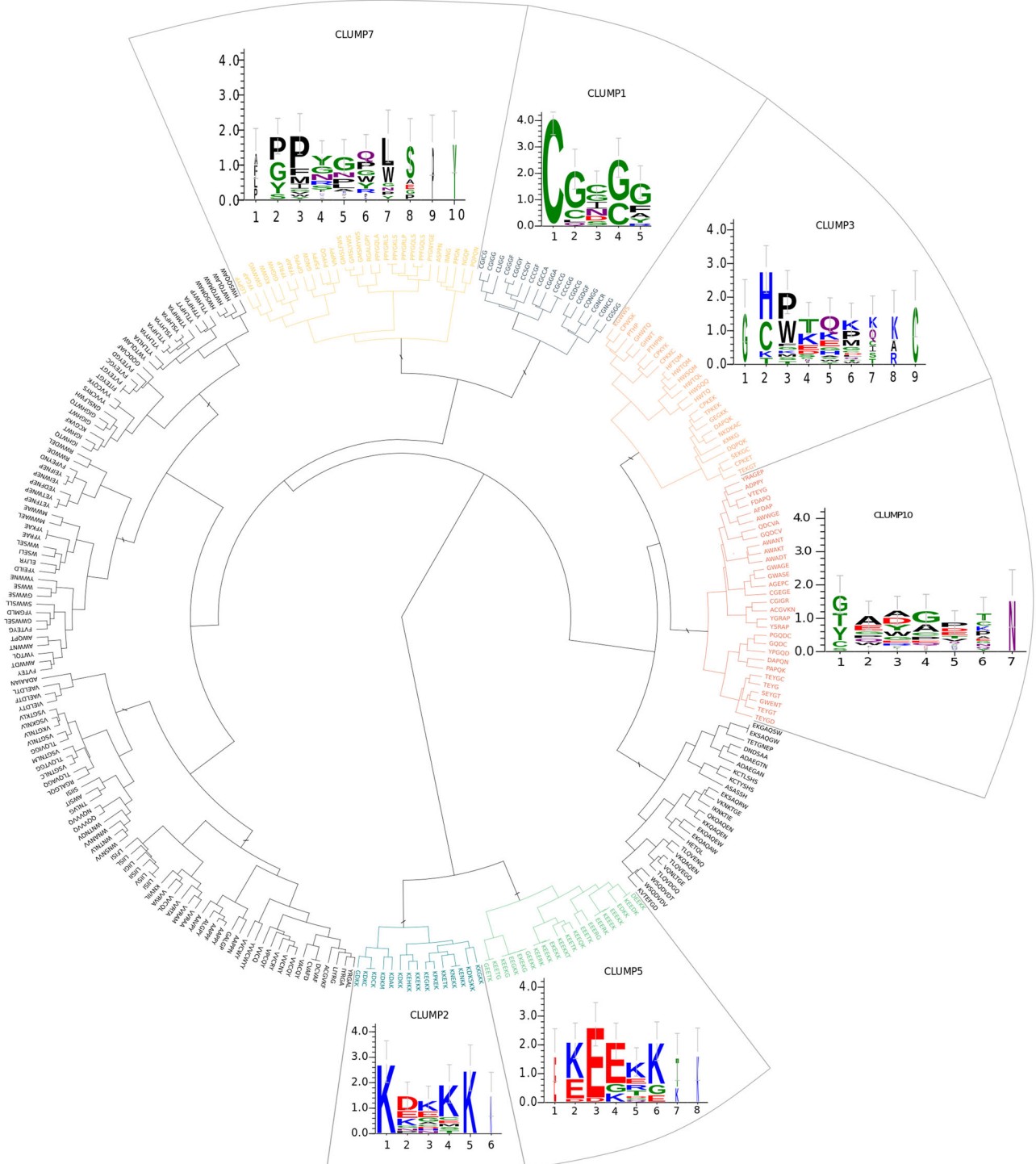

**Fig. 3 | Dendrogram of CLUMPs in Plant Parasitic Nematodes (PPNs).** 11 CLUMPs produced by MOnSTER (indicated with "/" sign). The coloured ones are those selected as best-scoring CLUMPs after MOnSTER-score calculation. Each best-scoring CLUMP is associated with the corresponding motif logo; alignment of motifs in each CLUMP is produced by PROMOCA and then WebLogo 3 is used to produce the image (the x-axis shows the AA position of the motif and the y-axis represents the log-transformed frequency of each AA in terms of bits of information).

the beginning of the sequence and consequently to the domain, while CLUMP10 mostly concentrates at the end of the domain, around 60–80% of the sequences (Supplementary Fig. 7). Examples of these genes in nematodes is poorly characterized and likely resulting from horizontal transfer[37,38]. Similarly, all 17 sequences presenting the co-occurrence of CLUMPs 3, 7, 10 also contain the cysteine-rich domain. Cysteine-rich

domain and CAP protein are known to be involved in the virulence of nematodes[39]. They are expressed in both plants and pathogens; in the latter, they are important for their virulence by suppressing the host's immune responses and promoting colonization. Interestingly, these sequences do not contain disordered regions or coil domains, consistently with unique conserved sandwich fold with a large central cavity of these kinds of proteins[40].

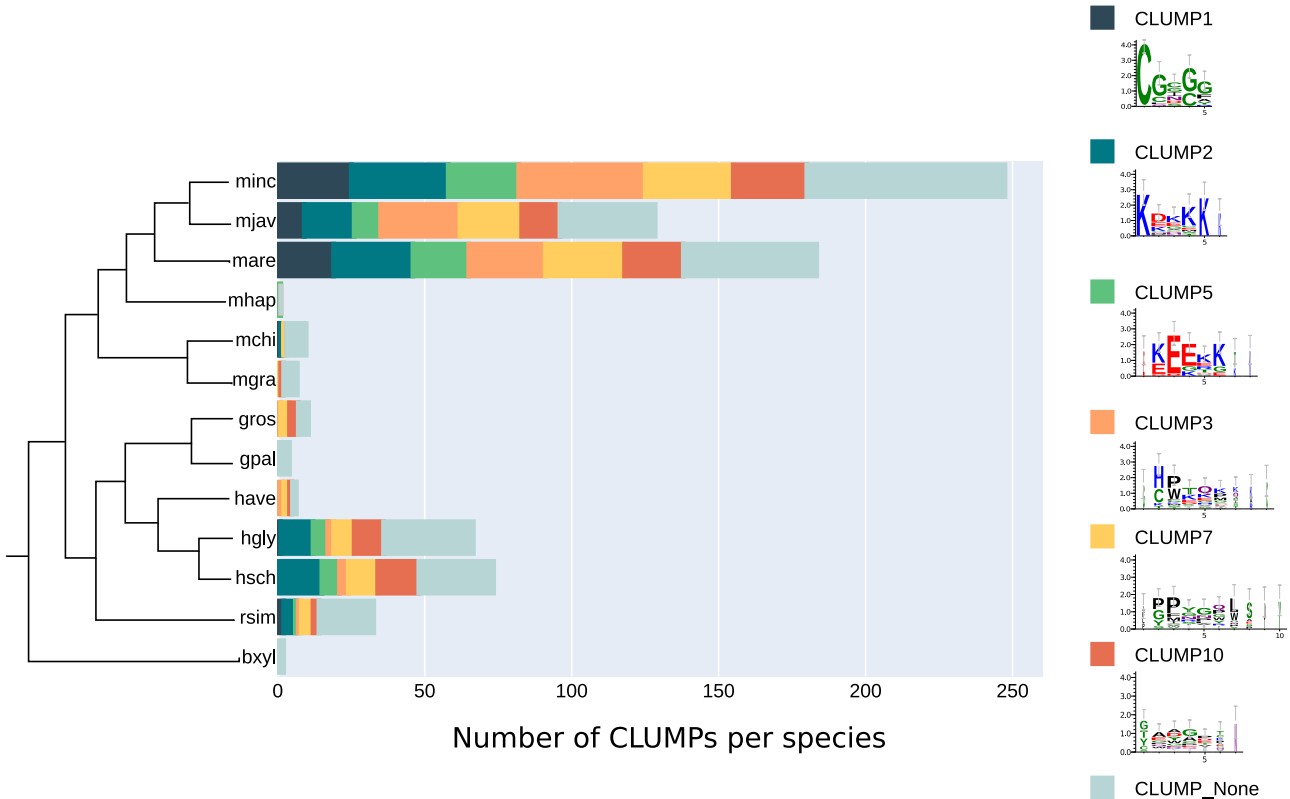

**Fig. 4 | Cardinality of CLUMPs-motifs in each PPN species considered.** The total number of motifs belonging to each selected CLUMP per PPN species accordingly to their phylogeny (minc: *Meloidogyne incognita*, mjav: *Meloidogyne javanica*, mare: *Meloidogyne arenaria*, mhap: *Meloidogyne hapla*, mchi: *Meloidogyne chitwoodi*, mgra: *Meloidogyne graminicola*, gros: *Globodera rostochiensis*, gpal: *Globodera pallida*, have: *Heterodera havenae*, hgly: *Heterodera glycines*, hsch: *Heterodera schachtii*, rsim: *Radopholus similis*, bxyl: *Bursaphelenchus xylophilus*).

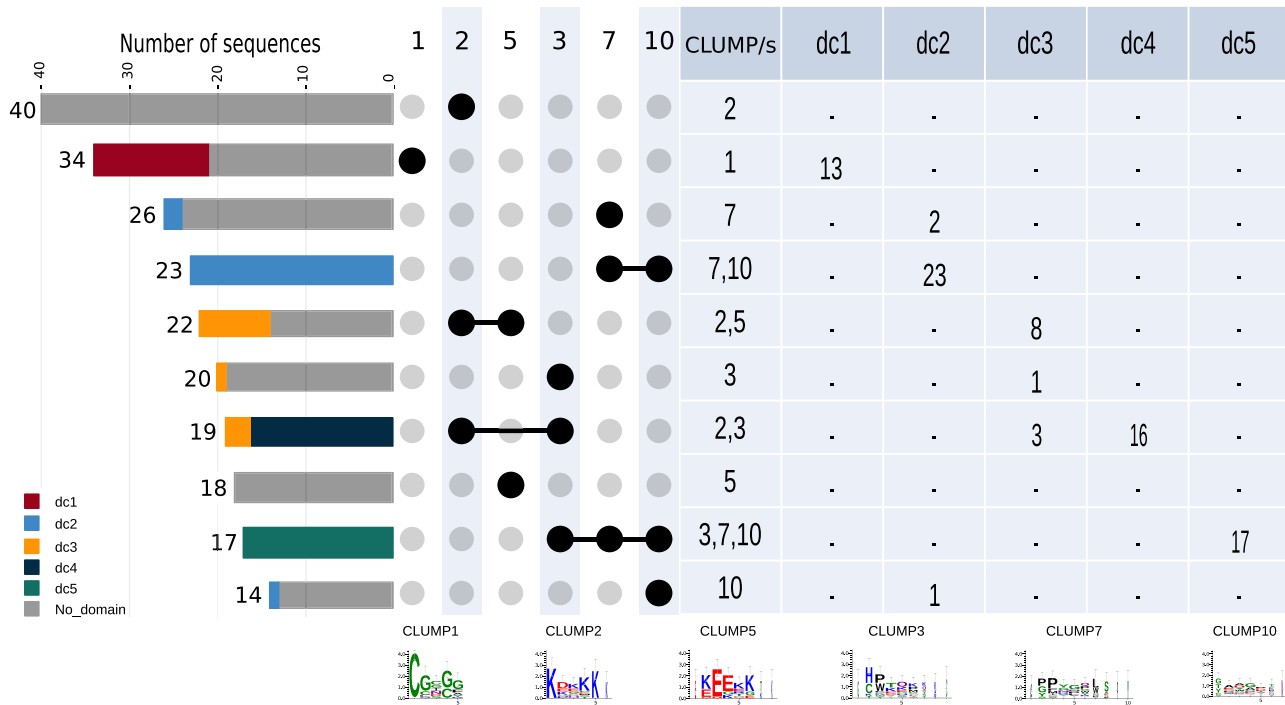

**Fig. 5 | Candidate parasitism proteins showing the presence of CLUMP/s associated with pathogenicity-related protein domain/s.** The table on the right shows the co-occurrence of CLUMP or CLUMPs with specific domain classes (dc); dc1, pectate lyase domain class, dc2, glycosyl hydrolase family 5 domain class, dc3 Stichodactyla toxin (ShK) domain class, dc4 14-3-3 family domain class and dc5, cysteine-rich domain class. The upset plot on the left represents the occurrences and co-occurrences of respective CLUMPs in the positive dataset, highlighting the sequences that also have an interesting protein domain following the table counts.

16 out of 19 sequences presenting co-occurrences of CLUMPs 2, 3 also have the 14-3-3 family domain, a eukaryotic-specific protein family with a general role in the signal transduction[41]. We also observe only one motif from CLUMP 2 in these sequences (KDKM) and 4 from CLUMP 3 (NKDKAC, KMKG, PTHPIR, PTHP). 13 out of 34 sequences bearing only CLUMP 1 also contain the pectate lyase domain. Of note, these sequences do not contain coiled or inordinate regions, and only seven show the presence of the SP. Pectate lyase enzymes in nematodes facilitate penetration in plant-cell walls made of pectin[42]. Numerous recent reports showed that these enzymes are produced in specialized nematode gland cells and secreted during the parasitism process. In the case of sedentary endo-parasitic nematodes, this occurs mainly during juvenile migration through the root tissue, when these enzymes play a crucial role in the maceration of the plant tissue facilitating the infection[43]. Finally, eight out of 22 sequences bear the co-occurrences of CLUMPs 2, 5 and the ShK domain. Although the exact biological function of the ShK domain remains unclear, previous reports have shown that this domain might be associated with immunosuppression[44,45].

Overall, these findings highlight that specific CLUMPs co-occurrences are associated with specific functional domains with roles in invasion and/or infection and might suggest different classes of candidate parasitism proteins cross-species.

## CLUMPs screening yielded the identification of a novel effector in *M. incognita* validated by in situ hybridization

To inspect whether the just identified CLUMPs could also help to find new effectors, we focused on the selection of a novel putative effector to validate experimentally. Thus, we selected all proteins of *Meloidogyne*

*incognita* proteome bearing the signal peptide for secreted proteins and no transmembrane domain. Then we screened these sequences and retrieved the ones containing at least one motif of the six selected CLUMPs. Among them, 23% contain at least one occurrence of motifs in CLUMP5 (Supplementary Table 3). Since this is the most abundant CLUMP in this species, we decided to focus on this one to identify a putative candidate to validate experimentally. By literature mining, we refined our list, by sorting out seven sequences that were already experimentally validated by previous studies (Supplementary Table 3). Then we filtered out any candidates having homologues in species other than root-knot nematodes and more than two gene copies to avoid dealing with multigene families according to[46]. Finally, among these eight putative effector sequences, we studied the pattern of expression of one candidate: *MiEFF72* (*Minc3s00056g02931*), by performing in situ hybridisation (ISH, see the "Methods" section). A specific signal was detected in the subventral oesophageal gland cells of pre-parasitic J2s after hybridisation with digoxigenin-labelled *MiEFF72* antisense probes (Fig. 6a). No signal was detected in pre-J2s with sense negative controls. *MiEFF72* fused to the C-terminus of GFP was transiently expressed in *N. benthamiana* leaf epidermis. GFP fluorescence was detected in the cytoplasm and in cytoplasmic vesicles (Fig. 6b). This finding suggests that *MiEFF72* be secreted and plays a role *in planta* in nematode parasitism.

Furthermore, we inspected the recently published *M. incognita* effectors not included in the positive dataset used to identify the CLUMPs, to check whether their sequences contain at least one of the six CLUMPs. Indeed, three effectors namely *MiTSPc* (*Minc3s00081g03887*)[47], *Minc3s00519g13668*[26] and *MiCE108* (*Minc3s00147g06033*)[48], bear at least one CLUMP/motif as reported in Fig. 6c. Altogether these results provide

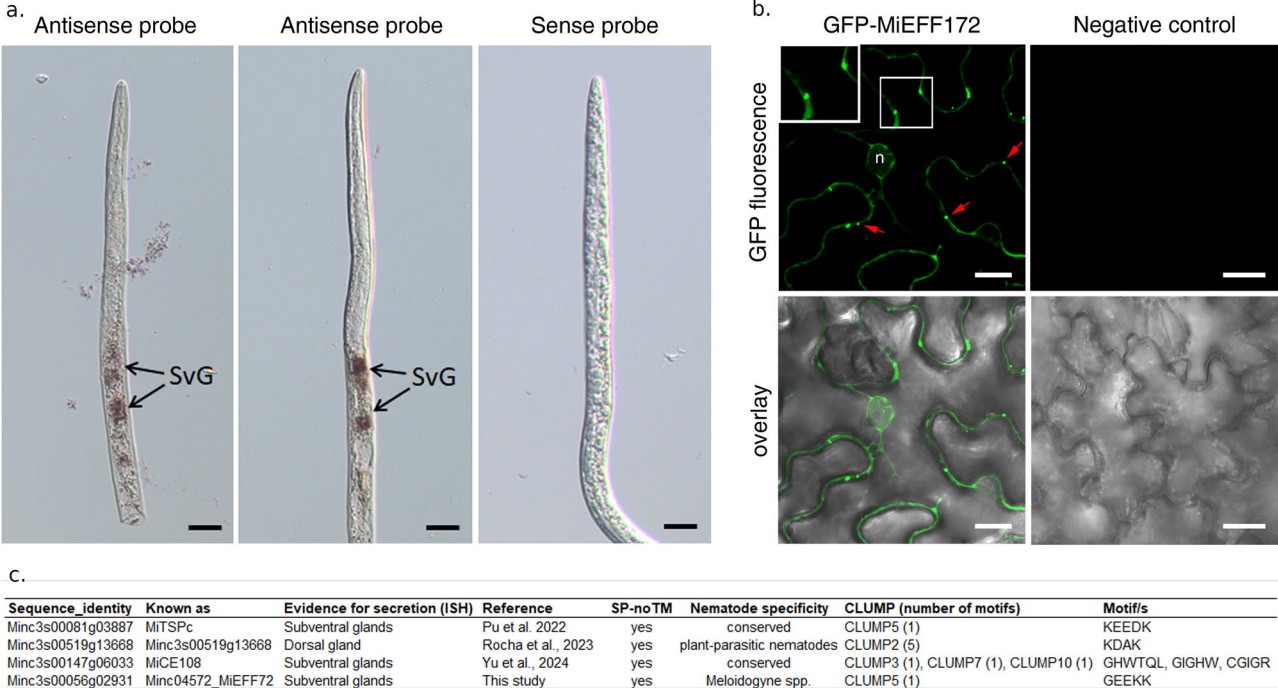

| Sequence_identity | Known as | Evidence for secretion (ISH) | Reference | SP-noTM | Nematode specificity | CLUMP (number of motifs) | Motif/s |
|---|---|---|---|---|---|---|---|
| Minc3s00081g03887 | MiTSPc | Subventral glands | Pu et al. 2022 | yes | conserved | CLUMP5 (1) | KEEDK |
| Minc3s00519g13668 | Minc3s00519g13668 | Dorsal gland | Rocha et al., 2023 | yes | plant-parasitic nematodes | CLUMP2 (5) | KDAK |
| Minc3s00147g06033 | MiCE108 | Subventral glands | Yu et al., 2024 | yes | conserved | CLUMP3 (1), CLUMP7 (1), CLUMP10 (1) | GHWTQL, GIGHW, CGIGR |
| Minc3s00056g02931 | Minc04572_MiEFF72 | Subventral glands | This study | yes | Meloidogyne spp. | CLUMP5 (1) | GEEKK |

**Fig. 6 | MiEFF72 is specifically expressed in the sub-ventral glands. a** In situ hybridisation showing EFF72 transcripts in the sub-ventral glands (SvG) of J2s of *M. incognita* (two left pictures). Sense probe for the MiEFF72 transcripts was used as a negative control (right picture). Bars = 20 μm. **b** MiEFF72 localised to the cytoplasm of plant cells and in cytoplasmic vesicles (red arrows and insert). The MiEFF72 sequence was fused to C-terminal end of the GFP and expressed in *N. benthamiana* leaves by agroinfiltration. Leaves infiltrated with water was used as negative control. Confocal GFP fluorescence images and overlay of differential interference contrast and fluorescence images are shown. Bars = 20 μm.

**c** Characteristics of recently published confirmed effectors of *M. incognita* with no relevant BLAST result or orthogroup with a positive dataset. Sequence ID, other common names for sequence ID, secretion localization based on in situ hybridization (ISH), reference publication, presence of signal peptide and absence of transmembrane domain (SP no TM), nematode specificity, presence of CLUMPs (number of motifs in each clump indicated in the "xn" format, where "n" is the number of repeats), motif sequence (if more than one motif is found in the sequence from the same CLUMP it is indicated in the "xn" format; if the motifs are from more than one CLUMP, they are indicated in the respective order of the previous column).

additional evidence to the validity of our method and support the utility of the identified CLUMPs to characterize specific properties of PPN effectors.

## Conclusions

This work is structured around three main aims: (1) the development of a specific method to cluster and score discriminant motifs of protein sequences called MOnSTER, (2) the validation of the MOnSTER results by applying it to identify CLUMPs specific to candidate parasitism protein sequences of oomycetes (3) the application of MOnSTER to protein sequences from plant-parasitic nematodes with unprecedented discriminant motifs detection.

The motif discovery tools used in this study yielded different sets of motifs at the sequence level, as expected because they use different methodologies. While MERCI uses a graph-based approach including amino acid's physicochemical properties, STREME considers the position weight matrix to build a generalized suffix tree combined with a statistical test. With the application on oomycetes and PPNs we have shown the utility of MOnSTER to group motifs based on physicochemical properties of the amino acids beyond sequence similarities.

The application of MOnSTER on oomycetes yielded the identification of five CLUMPs corresponding to the well-known effector-related motifs like RxLR-dEER and LxLFLAK-HVLVxxP motifs in oomycetes. This demonstrated that the novel scoring method introduced by MOnSTER is a good parameter with which to calculate CLUMP specificity for effector protein sequences. When applied to the nematodes candidate parasitism protein, MOnSTER found six CLUMPs, not previously characterized. The main advantage of MOnSTER is that the definition of CLUMPs allowed us to reduce the degeneration of 265 and 269 motifs (oomycetes and nematodes, respectively), to 11 CLUMPs. Candidate parasitism protein sequences of both pathogens show some common characteristics. Indeed, selected CLUMPs-motifs are present in about 70% of the input proteins for oomycetes and 60% in PPN compared to 15% and 5% in the negative dataset proteins, respectively. Furthermore, around 30% of candidate parasitism protein sequences have co-occurring CLUMPs, in contrast with less than 1% of the negative dataset sequences, in both applications. The main difference between candidate parasitism protein-specific motifs of the two pathogens is the positional preference: the beginning of the sequence for oomycetes and the central C-terminal for PPNs. This highlights MOnSTER ability to cluster motifs specifically relevant for candidate parasitism protein sequences without privileging any portion of the sequence, like other motif discovery tools.

Concerning the identified motifs for PPN's candidate parasitism proteins, we observed that the pattern of occurrences and co-occurrences of CLUMPs in candidate parasitism protein sequences is associated with specific functional domains and might suggest the existence of different classes of candidate parasitism proteins. Importantly we did not observe any species-related preferences thus implying the generality of these results.

An inherent circularity problem in methods identifying enriched motifs in a dataset of interest is that the positive dataset itself might have been constructed based on the presence of some previously known motifs. However, in the present case, we included experimentally validated effectors, some of which do not contain canonical motifs or are not being initially screened via identification of specific motifs or domains, which partially alleviates the problem.

In conclusion, MOnSTER quantifies the motifs and sequence properties in each dataset provided, thus allowing a wide application to other protein classes. Since the MOnSTER score considers the physicochemical properties and occurrences of motifs in CLUMPs concerning the protein sequences provided, it works without the need for a reference dataset. Furthermore, the MOnSTER scores are normalized values, therefore, allowing direct comparison between different studies.

Our results highlighted that MOnSTER is a powerful method to cluster and score discriminant motifs in protein sequences according to their physicochemical properties and pattern of occurrences. To the best of our knowledge, this is the first tool providing motif clusterization based on physicochemical properties. It is also a tool that can be easily used on any set of protein sequences and a list of motifs. Therefore, by constructing a dataset of positive candidate parasitism protein sequences and a negative dataset, MOnSTER can also be used to identify CLUMPs characteristics of fungal or bacterial candidate parasitism proteins. As such, MOnSTER can be included in any pipeline needing motif calling and will be of great use to accelerate both computational and experimental studies relating to protein motif discovery.

## Methods

### Oomycetes

We used proteins from five oomycetes species to create the input datasets for MOnSTER, namely *Phytophthora infestans*, *Phytophthora sojae*, *Phytophthora ramorum*, *Hyaloperonospora arabidopsidis* and *Bremia lactucae*.

The positive dataset consists of 1743 effector proteins belonging to the aforementioned oomycetes obtained from a concatenation of proteins selected from the PHI-base database (v4.14)[49], Uniprot (release 2023_02)[50], and the work of Haas et al.[33], in which they have manually curated the annotations of the proteins. Since the proteins come from different sources, we used CD-HIT (v4.8.1)[51] with the parameters in the Tools configuration paragraph, to filter out identical protein sequences. A total of 1283 proteins are annotated as RxLR effectors, 377 as Crinkler effectors and the last 83 sequences are proteins with no previously identified motif and known to be involved in the host-pathogen interaction.

Proteins in the negative dataset derive all from Uniprot (release 2023_02) and from the oomycetes species cited before being filtered from proteins included in the positive dataset and for evident effector-related annotations. Due to the large amount of non-effector proteins remaining from the filtering, we firstly used CD-HIT to reduce protein sequence redundancy and then, to also reduce the unbalance of the final dataset, we refined the selection, taking only the representative sequences of the orthogroups found with Orthofinder (v2.5.4)[52]. In total 3009 non effector proteins are included in the negative dataset.

The last input file consists of a list of motifs identified as enriched in the sequences of the positive dataset compared to the sequences of the negative one. We used MERCI and STREME (v5.5.1)[53], with parameters detailed in the Tools configuration paragraph. We imposed different lengths for motif prediction to be inclusive but more stringent on the motifs in which we are interested. STREME's output is a list of motifs. Hence, we used the tool FIMO (v5.5.1)[54], with default parameters to extract 246 degenerated motifs from the 4524 different motifs.

We obtained the following numbers of non-redundant motifs: 19 with MERCI and 246 with STREME. Then, we removed the identical motifs and created a single non-redundant list containing all the motifs in the same format, which resulted in 265 different motifs.

### Plant parasitic nematodes (PPNs)

The positive dataset contains candidate parasitism proteins selected to be likely secreted by PPNs in their plant host and belonging to 13 species (*Meloidogyne incognita*, *Meloidogyne javanica*, *Meloidogyne arenaria*, *Meloidogyne hapla*, *Meloidogyne chitwoodi*, *Meloidogyne graminicola*, *Globodera rostochiensis*, *Globodera pallida*, *Heterodera havenae*, *Heterodera glycines*, *Heterodera schachtii*, *Radopholus similis*, *Bursaphelenchus xylophilus*). We collected candidate parasitism protein from literature mining. More precisely we considered as candidate parasitism proteins those proteins for which in-situ hybridization experiments showed that the corresponding transcript is present in nematode secretory glands (dorsal or subventral), implying that these proteins are likely secreted by the nematodes into the host plant. The literature mining led to the extraction of 163 proteins from NCBI GeneBank thanks to the NCBI 'entrez' API. We also manually extracted 41 sequences from the publications' core text and supplementary information. In addition, we downloaded 41 sequences from WormBase ParaSite ([www.parasite.wormbase.org](www.parasite.wormbase.org), vWBPS17-WS282[55,56]),

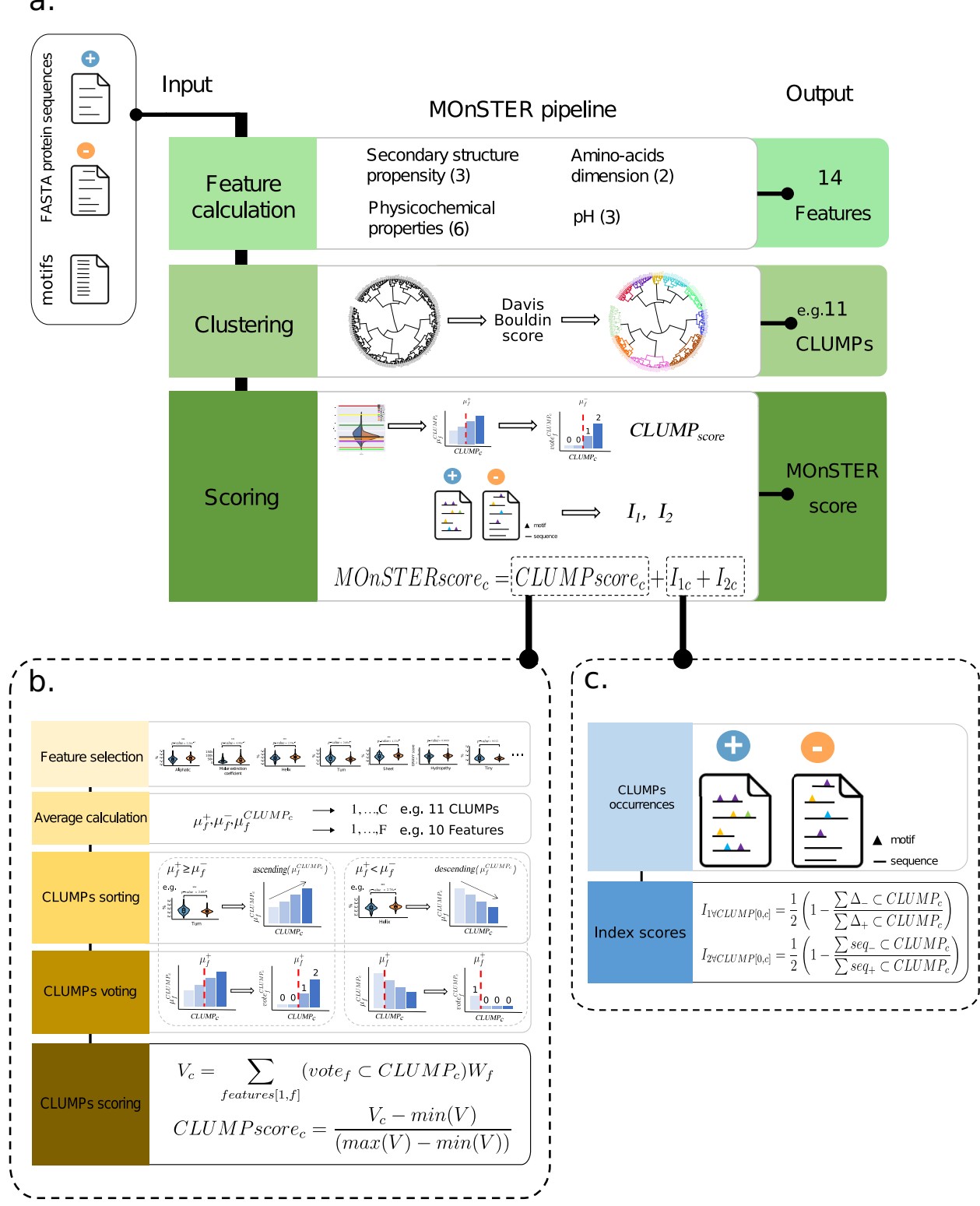

**Fig. 7 | MOnSTER pipeline scheme. a** MOnSTER pipeline is composed of three steps. It takes two FASTA protein sequences datasets (positive and negative) and a list of predicted motifs (enriched in the positive dataset) as input. The output is a list of CLUMPs and an associated MOnSTER score. The MOnSTER score is constituted by: **b** CLUMP$_{score}$ calculation. **c** Two occurrences Indexes.

and eight sequences from nematode.net[57]. In total, we obtained 229 candidate parasitism proteins. We extended the positive dataset with proteins that are non-redundant homologues of the previous candidate parasitism proteins in PPN proteomes. We first used cd-hit-2D with parameters in the

"Tools configuration" section, to cluster sequences from PPNs proteomes and candidate parasitism proteins[58]. We then pooled all the candidate parasitism proteins from closely related *Meloidogyne* species (e.g., *M. incognita*, *M. javanica* and *M. arenaria*) and scanned each corresponding

proteome with this multi-species set of sequences using cd-hit. Since the remaining species are genetically distinct, we then scanned each proteome with the relative set of candidate parasitism proteins, except for *H. havenae* and *M. chitwoodi* for which no proteomes were currently available. We merged the two sets of selected candidate parasitism proteins, and we performed CD-HIT intra- and inter-species to reduce dataset redundancy (parameters in the "Tools configuration" section), retaining only sequences having more than 1% divergence and aligning on more than 80% of their length (the longest sequence from each cluster was kept). The final positive dataset includes 546 candidate parasitism proteins from 13 species.

The negative dataset is composed of 3849 protein sequences that we obtained by selecting genes widely conserved across the nematode tree of life and close outgroup species, including many species that are non-parasites. Specifically, we filtered the results from a previous analysis[46] and only retained genes from orthogroups (i) conserved in more than 90% (62/64) of the analysed species including two tardigrade species (outgroups), and (ii) presenting <10 genes/species/orthogroups to avoid multigenic families, which would lead to overrepresentation of some proteins. To remove the redundancy, we used the same strategy as for the positive dataset (cd-hit-2D first and then CD-HIT).

Using the aforementioned software in the same configuration, we obtained the following numbers of non-redundant motifs: 40 with MERCI and 229 with STREME applying FIMO. In total, we obtained 269 different motifs.

All datasets are available at https://github.com/Plant-Net/MOnSTER_PROMOCA.git[31] and in Supplementary Data 2.1-2.2 and 3.1-3.2.

## Tools configuration

cd-hit-2D is used with the following configuration: -s2 0 -c 0.90 -g1 -aL 0.30 -aS 0.30. Where, s2, -c, -g parameter values are the default ones. -aL and -aS values are set so each sequence of a pair must cover at least 30% of the other one; CD-HIT: s = 0.8, c = 0.99, g = 1, aL = 0.80, aS=0.80; STREME, version 5.5.1, accessible at https://meme-suite.org/meme/doc/download.html, is used with the following parameters: -minw 3 -maxw 5; -minw 3 -maxw 7; MERCI, accessible at http://dtai-static.cs.kuleuven.be/ml/systems/MERCI/MERCI.zip, is used with parameter: -l 5 -fp 20; -l 7 -fp 20; -l 10 -fp 20.

## MOnSTER pipeline

The MOnSTER (MOtifs of cluSTERs) pipeline is composed of three main steps as described in Fig. 7 and in the following sections.

## MOnSTER pipeline—feature calculation

The first step of the pipeline concerns the calculation of parameters that describe protein sequences (Fig. 7a). To allow an easy calculation of the features on any dataset, we calculated the sequence length and used *ProteinAnalysis* class from the *Bio.SeqUtils.ProtParam*, a python sub-package to select 13 additional features based on individual AA properties, belonging to 4 categories:

- secondary structure propensity 'helix' (V, I, Y, F, W, L), 'turn' (N, P, G, S), and 'sheet' (E, M, A, L).
- amino-acids dimensions ('tiny' (A, C, G, S, T) and 'small' (A, C, F, G, I, L, M, P, V, W, Y)).
- pH ('basic' (H, K, R), 'acid' (B, D, E), and 'charged' (H, K, R, B, D, E)).
- physicochemical properties ('hydropathy-score', 'polar' (D, E, H, K, N, Q, R, S, T), 'non-polar' (A, C, F, G, I L, M, P, V, W, Y), 'aromatic' (F, H, W, Y), and 'aliphatic' (A, I, L, V)).

For each of the 13 sub-categories, we calculated the cumulative percentage of the associated AA in each sequence as the value for each corresponding feature. The hydropathy score is the equivalent of the GRAVY (grand average of hydropathy) value, introduced by Kyte and Doolittle[59]. Accordingly, the score is obtained by the hydropathy value of each sequence residue normalized by its length. The length of each motif is also used as an additive feature, leading to 14 total features.

We performed feature calculations on the positive and negative datasets and the list of motifs. At the end of this step, we obtained three tables of features, one for each of the input datasets (positive, negative datasets and the list of motifs).

## MOnSTER pipeline—Clustering

This step allowed to cluster motifs based on their properties described by the 13 features. To make the features comparable to each other, we performed data normalization by using the *StandardScaler* method from *sklearn.preprocessing*[60]. This normalization consists of the removal of the mean and the scaling to unit variance.

Then, we performed a hierarchical clustering of the motifs using the Euclidian distance. We then divided the resulting tree into clusters of motifs of proteins (CLUMPs) selecting the threshold distance that minimized the Davies–Bouldin score[61].

For each CLUMP, we removed the redundant motifs. Briefly, we identified motifs that shared a core sequence (for example: 'HWT in HWTQ' and 'GHWTQ'), and we only retained the cores (for instance: "HWT") in the CLUMPs.

## MOnSTER pipeline—Scoring

The final objective is to identify the CLUMP(s) with the highest discriminative power concerning the positive dataset. Thus, we conceived a new score called the MOnSTER score, to rank the CLUMPs by their discriminative power.

The MOnSTER score is composed of three parts: the CLUMP score and two modified versions of the Jaccard index.

## MOnSTER pipeline—*CLUMP* score

This score considers the AA composition of the motifs belonging to each CLUMP concerning the preferences of the sequences of the positive dataset. The procedure that we implemented to calculate this score is shown in Fig. 7b.

**Feature selection**. We used the Mann–Whitney test to identify the features whose values were significantly different between the positive and negative datasets. We only retained the statistically significant features, with a *p*-value < 0.05. Then, we assigned them a score, by calculating $-\text{Log}(p\text{-value})$ of each feature. We will refer to it as the 'feature weight' hereafter.

**Average calculation**. For each of the selected features (ranging from one to *f*), we calculated the average value for the positive dataset, the negative dataset, and each CLUMP (ranging from zero to *c*). We will refer to these values with the notation: $\mu_f^+$, $\mu_f^-$ and $\mu_f^{\text{CLUMP}_c}$, respectively.

**CLUMPs sorting**. We compared the averages of the positive and negative datasets for each feature and sorted CLUMPs accordingly.

Thus, if the $\mu_f^+ \geq \mu_f^-$, the CLUMPs averages would be sorted in ascending order.

Otherwise ($\mu_f^+ < \mu_f^-$), CLUMPs averages would be sorted in descending order.

**CLUMPs voting**. For each feature, and each CLUMP, we divided the CLUMP into two groups according to the following statements:

If $\mu_f^+ \geq \mu_f^-$: CLUMPs with $\mu_f^{\text{CLUMP}_c} \geq \mu_f^+$ have a vote from 1 to the number of CLUMPs with an increment of 1, otherwise the score is set to 0.

If $\mu_f^+ < \mu_f^-$: CLUMPs with $\mu_f^{\text{CLUMP}_c} < \mu_f^+$ the vote attributed goes from 1 to the number of CLUMPs, otherwise it is 0.

**CLUMPs scoring**. For each CLUMP (ranging from zero to *c*), for each feature (ranging from one to *f*), we multiplied the feature-vote by the 'feature weight' ($W_f$) and summed-up to obtain a CLUMP-vote. Then we scaled each CLUMP-vote to a range from 0 to 1 using the following formula:

$$\text{CLUMPscore}_c = \frac{V_c - \min(V)}{(\max(V) - \min(V))}$$

where

$V$ is the list of CLUMPs votes and $V_c$ is calculated as

$$V_c = \sum_{\text{features}[1,f]} \left( \text{vote}_f \subset \text{CLUMP}_c \right) W_f$$

## MOnSTER pipeline—Occurrences indexes

The two indexes respectively consider: (i) the occurrences of the motifs, for each CLUMP, in the positive dataset compared to the negative and (ii) the number of positive sequences containing the motifs in each CLUMP concerning the negatives (Fig. 7c).

**CLUMPs occurrences.** We calculated the occurrences of the motifs in each CLUMPs in the two datasets (positive and negative).

**I's scores.** We propose two ways to calculate the dissimilarity between two sets that will be called $I_1$ and $I_2$ hereafter.

To obtain $I_1$, we calculated the number of occurrences of the motifs for each CLUMP (ranging from zero to $c$) in the negative dataset over the number of occurrences of the motifs of the same CLUMP in the positive dataset, using the following equation:

$$I_{1\forall\text{CLUMP}[0,c]} = \frac{1}{2}\left( 1 - \frac{\sum \triangle_- \subset \text{CLUMP}_c}{\sum \triangle_+ \subset \text{CLUMP}_c} \right)$$

where

$\triangle_-$ and $\triangle_+$ the number of occurrences of the motifs of the CLUMP in the negative or in the positive dataset, respectively

To obtain $I_2$, for each CLUMP (ranging from zero to $c$), we calculated the number of sequences of the negative dataset that contain at least a motif of the CLUMP, over the number of sequences of the positive dataset that contain at least a motif of the same CLUMP, accordingly to the following formula:

$$I_{2\forall\text{CLUMP}[0,c]} = \frac{1}{2}\left( 1 - \frac{\sum \text{seq}_- \subset \text{CLUMP}_c}{\sum \text{seq}_+ \subset \text{CLUMP}_c} \right)$$

where

$\text{seq}_-$ is the number of sequences of the negative dataset containing at least a motif of the CLUMP.

$\text{seq}_+$ is the number of sequences of the positive dataset containing at least a motif of the CLUMP.

The ½ factor is applied to have values between 0 and 0.5 for each Index to have equal weight in the final score, and (1–Index) is to consider the degree of dissimilarity rather than similarity.

## MOnSTER pipeline—MOnSTER score

The MOnSTER score, for each CLUMP (from zero to $c$), is the sum of the corresponding CLUMP score, and the two I indexes:

$$\text{MOnSTERscore}_c = \text{CLUMPscore}_c + I_{1c} + I_{2c}$$

## PRO-MOCA: a method to create motif logo of CLUMPs

To create motif logos for each CLUMP, we developed PRO-MOCA (PROtein-MOtifs Characteristics Aligner), which aligns protein motifs based on the characteristics of the amino acids as shown in Supplementary Fig. 8. The first step is to define the alphabets associated with each characteristic that can be used to represent the motifs (Supplementary Fig. 8a). We have defined four alphabets, namely: "chemical", "hydrophobicity", "charge", "secondary structure propensity".

These alphabets are easily modifiable and other alphabets can be included. Different CLUMP logos can be obtained depending on the alphabet chosen. Secondly, PRO-MOCA uses the selected alphabet to translate the AA sequences of each motif in a CLUMP in the defined alphabet (Supplementary Fig. 8b). The third step is the alignment (Supplementary Fig. 8c). Briefly, PRO-MOCA screens the translated motif sequences of a CLUMP looking for a "summit position" with the highest frequency of the same "letter" of the just defined alphabet. Once this position is identified, all motifs are aligned accordingly (Supplementary Fig. 8d). Since the motifs of a CLUMP can have different lengths, after the alignment, PRO-MOCA calculates the number of gaps to add at the extremities to make all motifs having the same length. Importantly, gaps are not allowed inside the motif sequences. The last step of the method is to re-translate the aligned motifs in the original AA sequences (Supplementary Fig. 8e). The alignment is ready to feed a programme to create logos. Here we used the tool Weblogo3[62].

## PPNs candidate parasitism protein domains mining analysis

To investigate the relationship between the selected CLUMPs and functional domains in candidate parasitism proteins, we first selected proteins from the positive datasets containing at least one occurrence of a selected CLUMP (311 proteins in total). Then we predicted the protein domains with InterProScan (v5.54-87.0)[30] with default parameters. From the results, we extracted the proteins containing the most frequent predicted domains and considered only entries coming from MobiDB-lite, Coils, CDD, PANTHER, Pfam and ProSitePatterns. Afterwards, we also predicted the presence of Signal Peptide (SP) (SignalP4.1[63]) and TransMembrane (TM) domain regions (TMHMM2.0[64]). We obtained 258 proteins having at least a CLUMP and one of the aforementioned predicted domains, SP or TM.

## In situ hybridisation (ISH) and N. benthamiana agroinfiltration

*M. incognita* strain "Morelos" was multiplied on tomato (*Solanum lycopersicum* cv. "Saint Pierre") growing in a growth chamber (25 °C, 16 h photoperiod). Freshly hatched J2s were collected and ISH performed as previously described[65,66]. The *M. incognita Minc3s00056g02931/MiEFF72* coding sequence (CDS) lacking the signal peptide for secretion was amplified by PCR with specific primers (EFF72_F: 5'-AAAAAG-CAGGCTTCACCATGAATACTGCTGACAAGACACAG-3' and EFF72_R: 5'- AGAAAGCTGGGTGTTAGAACAAAGCTCGCACTGC-3') and inserted into the pDON207 entry vector. Antisense probe was amplified using EFF72_R from the entry vector. Sense probe was amplified using EFF72_F and used as a negative control. Images were obtained with a microscope (Axioplan2, Zeiss, Germany).

The *M. incognita MiEFF72* CDS lacking the signal peptide was recombined in pK7FGW2 (P35S:eGFP-MiEFF72) with Gateway technology (Invitrogen). The construct was sequenced (GATC Biotech) and transferred into *Agrobacterium tumefaciens* strain GV3101. Transient expression was achieved by infiltrating *N. benthamiana* leaves with *A. tumefaciens* GV3101 strain harbouring the GFP-fusion construct, as previously described[67]. Leaves were imaged 48 h after agroinfiltration, with an inverted confocal microscope (LSM880, Zeiss, Germany) equipped with an Argon ion and HeNe laser as the excitation source. GFP emission was detected selectively with a 505–530 nm band-pass emission filter.

## Statistics and reproducibility

To obtain motifs enriched in the positive dataset for both applications, we used the internal calculation of STREME and varied the minimal frequency threshold of motifs in positive protein sequences, tuning the -fp parameter of MERCI.

For feature selection needed for the CLUMP score calculation, we used the Python library scipy[68] to perform the Mann–Whitney test on both positive datasets (1743 proteins for oomycetes and 546 for PPNs) and negative datasets (3009 proteins for oomycetes and 3849 for PPNs). The statistically significant threshold is a $p$-value < 0.05.

For each ISH experiment (each probe), 10,000 *Meloidogyne incognita* juvenile (J2) were used as described in Jaouannet et al.[66]. This number means that at least a hundred nematodes can be observed at the end of the experiment, despite losses during the various treatments. For *Nicotiana benthamiana* agroinfiltration, a minimum of three leaves were agroinfiltrated and observed for the construct (or water control) in each experiment in order to take into account the possible variability of expression between leaves. ISH with the antisense probe was carried out three times independently and 30 pictures were taken; ISH with the sense negative probe were carried once and 12 pictures were taken; Agroinfiltrations with the GFP fusion or negative control (water) were carried out three times independently and 25 and 13 pictures were taken, respectively. All attempts at replication were successful, i.e. signals observed for ISH antisense probes or GFP fusion for agroinfiltration, or not in the case of negative controls (sense probe or water infiltration, respectively).

## Reporting summary

Further information on research design is available in the Nature Portfolio Reporting Summary linked to this article.

## Data availability

Input and output data of MOnSTER are available at the public GitHub and Zenodo repositories: https://github.com/Plant-Net/MOnSTER_PROMOCA.git, https://doi.org/10.5281/zenodo.11368310[31]. All other data remain available from the corresponding author on reasonable request. Source data for the graph in Fig. 4 can be found in Supplementary Data 1.4.

## Code availability

The source code is available at the public GitHub repository https://github.com/Plant-Net/MOnSTER_PROMOCA.git[31].

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

## Acknowledgements

This work was supported by the French government, through the UCA JEDI Investments in the Future project managed by the National Research Agency (ANR) under reference number ANR-15-IDEX-01. Microscopy work was performed at the SPIBOC imaging facility of Institut Sophia Agrobiotech, and we thank Dr Olivier Pierre for his availability. The funders had no role in study design, data collection and analysis, decision to publish, or preparation of the manuscript.

## Author contributions

G.C.: Methodology, software, validation, formal analysis, writing—original draft, visualization. P.P.: Methodology, software, writing—original draft. J.C.: Investigation, resources. D.K.: Software, resources. H.S.: Writing—review & editing, supervision. A.C.: Writing—review & editing, supervision. M.Q.: Investigation, resources, writing—review & editing, supervision. B.F.: Investigation, resources, writing—review & editing, supervision. E.G.J.D.: Conceptualization, methodology, writing—review & editing, supervision. S.B.: Conceptualization, methodology, writing—original draft, writing—review & editing, supervision, project administration.

## Competing interests

The authors declare no competing interests. Hannes Schuler is an Editorial Board Member for Communications Biology, but was not involved in the editorial review of, or in the decision to publish this article.
