## [Peer Review File · Communications Biology]

Reviewers' comments:

Reviewer #1 (Remarks to the Author):

In this manuscript, the authors developed MOnSTER to identify clusters of motifs in protein sequences. They apply the proposed methods in oomycete effectors and plant-parasitic nematode effectors.

Considering the proposed method is based on two existing motif discovery tools, the current work represents a minor yet significant contribution. The clustering may reduce the number of motifs in effectors, which allows biologists to know the sequence patterns within a protein family more conveniently. In general, the manuscript writing is fairly clear. I have only some minor comments.

1) Are there any existing motif clustering methods available to the community? If yes, the authors should discuss and compare the results from different clustering methods.

2) To my knowledge, the experimentally verified oomycete effectors are quite limited. I am wondering if the effectors selected from public databases have experimental evidence?

3) Some terms should be used with the same format. For example, CD-HIT and cd-hit are simultaneously used in this manuscript. The authors should check this issue throughout the whole manuscript.

4) It seems that the motifs identified from MERCI and STREME are quite different. The authors should comment on this issue and justify the joint use of these two motif discovery methods. In a further step, the authors should discuss the final MOnSTER results' relatedness to the motifs from MERCI and STREME.

Reviewer #2 (Remarks to the Author):

the authors describe a tool (Monster) which can identify short amino acid motifs like those of RXLR and CRN oomycete effectors, but unlike most methods (pfam/intepro) there is a post-matching clustering step (CLUMPs), which might offer additional insight during effector discovery

while mostly well-written the manuscript had a few grammatical errors/typos throughout,

e.g. abstract, line 36, pathogens  pathogen

acronym PPN also used in abstract but not defined

a thorough revision of the whole text is needed in this regard, otherwise the paper was interesting and suitable for publication after revisions below

they demonstrate its utility by benchmarking monster with a positive and negative dataset across multiple nematode species, and experimentally validate the subcellular localisation of one effector candidate

the positional analysis is also interesting as a starting point for incrementally understanding pathogen protein module/domain structure vs function

the taxonomic analysis vs CLUMPs is also an interesting result for the PPNs and perhaps a version of the oomycete supplementary figure could be added to the main text

paragraph formatting of data availability section is justified and should be edited

this comment relates to all figures with dc/CLUMP IDs - include the CLUMP motif/pattern in legend as this will be more informative - otherwise the reader must look up each CLUMP id from a supplementary table

Reviewer #3 (Remarks to the Author):

I liked this manuscript. It read well and represents a solid addition to the set of tools and techniques for analysing functional transcription factor motifs. I did think that there was a circularity to the background sets used, though I couldn't think of a clear alternative that would alleviate the problem. I encourage the authors to discuss this issue a little in the conclusions section.

I thought the method was not as well described in text as it was in the very clear code implementation provided. Could the authors provide extra clarity and detail about the actual feature selection e.g the use of cumulative percent as a feature that isn't mentioned in the text of the manuscript.

Reviewers' comments:

Reviewer #1 (Remarks to the Author):

In this manuscript, the authors developed MOnSTER to identify clusters of motifs in protein sequences. They apply the proposed methods in oomycete effectors and plant-parasitic nematode effectors. Considering the proposed method is based on two existing motif discovery tools, the current work represents a minor yet significant contribution. The clustering may reduce the number of motifs in effectors, which allows biologists to know the sequence patterns within a protein family more conveniently. In general, the manuscript writing is fairly clear. I have only some minor comments.

1) Are there any existing motif clustering methods available to the community? If yes, the authors should discuss and compare the results from different clustering methods.

To the best of our knowledge, there are no tools available for clustering protein motifs. We have added this consideration in the Conclusion section, please see lines 568-570.

2) To my knowledge, the experimentally verified oomycete effectors are quite limited. I am wondering if the effectors selected from public databases have experimental evidence?

In our positive dataset of oomycete effectors there are 102 experimentally validated proteins including 98 PHI-base proteins (PHI-base entries are supported by strong experimental evidence from a peer-reviewed publications), and four CRN effectors from Haas et al 2009. These information are in Supplementary table 1.1.

3) Some terms should be used with the same format. For example, CD-HIT and cd-hit are simultaneously used in this manuscript. The authors should check this issue throughout the whole manuscript.

We thank the Reviewer for this suggestion, we have now revised this, please see line 164.

4) It seems that the motifs identified from MERCI and STREME are huge different. The authors should comment this issue and justify the jointly use of these two motif discovery methods. In a further step, the authors should discuss the final MOnSTER results' relatedness to the motifs from MERCI and STREME.

Indeed, we agree with the Reviewer that it is important to emphasize the difference in the found motifs by the two tools and therefore the utility of MOnSTER to clusterize them. We discuss this matter in the Conclusion section, please see lines 530-535.

Reviewer #2 (Remarks to the Author):

the authors describe a tool (Monster) which can identify short amino acid motifs like those of RXLR and CRN oomycete effectors, but unlike most methods (pfam/intepro) there is a post-matching clustering step (CLUMPs), which might offer additional insight during effector discovery

while mostly well-written the manuscript had a few grammatical errors/typos throughout, e.g. abstract, line 36, pathogens  pathogen

acronym PPN also used in abstract but not defined

a thorough revision of the whole text is needed in this regard, otherwise the paper was interesting and suitable for publication after revisions below

We thank the reviewer for her/his appreciation for our work. We have now corrected those errors and others through the manuscript.

they demonstrate its utility by benchmarking monster with a positive and negative dataset across multiple nematode species, and experimentally validate the subcellular localisation of one effector candidate

the positional analysis is also interesting as a starting point for incrementally understanding pathogen protein module/domain structure vs function

the taxonomic analysis vs CLUMPs is also an interesting result for the PPNs and perhaps a version of the oomycete supplementary figure could be added to the main text

We thank the reviewer for the positive comments regarding our work. Following reviewer's suggestion, we have moved supplementary figure 2 in the main text as figure 3.

paragraph formatting of data availability section is justified and should be edited

We have edited that, please see lines 577-579.

this comment relates to all figures with dc/CLUMP IDs - include the CLUMP motif/pattern in legend as this will be more informative - otherwise the reader must look up each CLUMP id from a supplementary table

We have added the CLUMP motif logos in all concerned figures in the main text and supplementary material.

Reviewer #3 (Remarks to the Author):

I liked this manuscript. It read well and represents a solid addition to the set of tools and techniques for analysing functional transcription factor motifs. I did think that there was a circularity to the background sets used, though I couldn't think of a clear alternative that would alleviate the problem. I encourage the authors to discuss this issue a little in the conclusions section.

We thank the reviewer for the positive comments about our methodology for enriched motifs discovery. We understand the concerns about the possible circularity of the problem.

Indeed, it is true that presence of specific motifs is part of the list of criteria used by biologist to identify candidate effectors to be further validated by biological experiments.

For example, in oomycetes, presence of an RXLR motif in the protein sequence is usually a criterion to select candidate effectors and then validate their translocation in plant cells. However, in our positive dataset we also included experimentally validated oomycete effectors which do not contain the classical RXLR motif.

In nematodes, one criterion classically used for the identification of candidate effector is selection of proteins that do have a predicted signal peptide for secretion (SP) and no predicted transmembrane (TM) region. Those candidate effectors are then further processed for experimental validation (confirmation of transcription in secretory glands and / or confirmation of presence of the protein in planta). However, for nematode as well, we included experimentally validated effectors from the literature that were not necessarily identified via the simple SP no TM criterion. Our positive dataset also includes validated effectors that were initially selected from direct proteomic experiments on nematode secretions or transcriptome of dissected secretory gland cells. Furthermore, the nematode negative dataset also includes proteins that have a SP and no TM. Therefore, in the case of nematodes the risk of circularity is minimal.

Overall, we can consider that this inherent circularity problem is partly compensated by the inclusion in the positive datasets of proteins not originally identified via presence of peptide motifs.

Accordingly, we have added a sentence to discuss this point in the manuscript, see lines 557-561.

I thought the method was not as well described in text as it was in the very clear code implementation provided. Could the authors provide extra clarity and detail about the actual feature selection e.g the use of cumulative percent as a feature that isn't mentioned in the text of the manuscript.

We agree with the Reviewer, and we have now improved the description of the features, please see lines 242-246.

REVIEWERS' COMMENTS:

Reviewer #1 (Remarks to the Author):

I appreciate the authors' efforts to address my previous comments.

Reviewer #2 (Remarks to the Author):

The manuscript is much improved, there were some minor typos in the newly added text which the authors should correct, as well as go over the text carefully for grammatical errors that may have been missed by reviewers:

line 85 should move (PPN) before the word effectors

530 sequences  sequence

867 sub-ventral